# Preparations for Galileo PRS in Poland

**DOI:** 10.3390/s23041770

**Published:** 2023-02-04

**Authors:** Krzysztof Bronk, Adam Lipka, Rafal Niski

**Affiliations:** National Institute of Telecommunications, 04-894 Warsaw, Poland

**Keywords:** GNSS, Galileo PRS, jamming, spoofing, GNSS threats detection system

## Abstract

This article discusses the increasing security risk for the Global Navigation Satellite System (GNSS) due to both unintentional and deliberate interference (attacks), which have gotten significantly worse in 2022 due to tense the international situation. The upcoming Galileo Public Regulated Service (PRS), which is more resilient and robust than initial GNSS open services, is one of the key solutions for that problem. The technical description of this service, aspects regarding its implementation in the EU and the role of designated governmental authorities in that process are extensively covered in the first sections of the article. The next relevant issue brought up in the paper is the PRS signals’ coexistence with amateur services operating within the same frequency resources, which have recently became a source of significant controversy in Europe. Finally, the article presents the Polish contribution to the Galileo PRS preparatory actions, covering the participation in two international R&D projects, the developed measurement station and initial results for the GNSS receiver’s jamming and spoofing resistance tests, as well as the concept of the Galileo PRS threats detection system.

## 1. Introduction

Many sectors of the European economy (transport, logistics, telecommunications, and energy) depend on precise location and timing. The European satellite navigation market is growing steadily, and consequently, the satellite navigation aspects are becoming more important, also due to the fact that the Galileo system is to become fully operational in 2023. The GNSS systems are not 100% reliable. There may be periods of outage or a significant decrease in their quality (e.g., as a result of unintentional or deliberate interference, such as jamming and spoofing), which may potentially result in life-threatening situations and, in some aspects, even threats to public safety. In the context of that, the following facts and incidents can be mentioned. First of all, the GPS is the US military system, and its availability for civilian users may be temporarily limited or even disabled in certain situations and areas. Secondly, during the well-known Galileo outage incident in July 2019, following a combination of several circumstances, the system was completely unavailable for seven consecutive days. Finally, the multiple reports of intentional and long-term interference with the GPS signals, which initially occurred in northern Europe territory, and since February 2022, has also been observed in eastern parts of Europe. 

In the context of increasing threats to the integrity and availability of GNSS systems, related to unintentional or deliberate activities, there is a need for a service more resilient and robust than the initial GNSS open services. The Galileo Public Regulated Service (PRS), dedicated for authorized European governmental users, is one of the most promising solutions. The PRS service will be encrypted and more resistant to typical spoofing and jamming attacks, and it will be crucial for the safety and security of the European Union as an alternative to open GNSS systems vulnerable to deliberate interferences.

## 2. Galileo Services and Signal Plan

Galileo is the European Global Navigation Satellite System (GNSS), providing worldwide radio signals for position, navigation and timing purposes. The European Commission is responsible for the Galileo program, and the European Union Agency for the Space Programme (EUSPA) is responsible for deploying the system and providing technical support for operational tasks. Galileo’s performance is gradually improving as the system’s space segment is getting close to its target parameters. When the system reaches the Full Operational Capability (FOC) phase, it will offer four high-performance services [1]:Open Service (OS), free of charge open service providing initial signals for positioning and timing purposes.High Accuracy Service (HAS), complementing the OS service and providing additional navigation signals and added-value services in a different frequency band.Public Regulated Service (PRS), restricted to government-authorized users and dedicated for sensitive applications requiring a high level of service continuity.Search and Rescue Service (SAR), Europe’s contribution to the international satellite-based search and rescue distress alert detection system COSPAS-SARSAT.

In order to facilitate the future improvement of the Galileo services’ quality, it is also planned that Galileo Open Service will provide Navigation Message Authentication, allowing users to utilize the authenticated data extracted from navigation messages. The HAS and PRS signals will be encrypted in order to restrict the access only to authorized and governmental users. PRS is primarily intended for use by European units, such as fire brigades, health services (ambulance), humanitarian aid, search and rescue, police, coastguard, border control, customs and civil protection units. Access to the PRS is controlled through operational and technical means, including governmental-grade encryption. Users not granted access to the service will be unable to extract any information from the signal. Moreover, the PRS service will be more resilient and more robust than the initial open services of Galileo and other GNSS systems. The PRS will ensure better continuity of service to authorized users when access to other navigation services may be degraded (resilience), and in cases of malicious interference, PRS will increase the likelihood of the continuous availability of the Signal-in-Space (robustness). The service’s end-to-end design ensures the protection and availability of the signal and its associated data flows. Due to the signal and system design, the PRS service will also be more resistant to typical spoofing and jamming attacks. 

Galileo satellites permanently transmit three independent CDMA and Right-Hand Circularly Polarized (RHCP) signals, denoted E1, E5 (sub-divided into E5a and E5b) and E6 [2]. These signals are transmitted on four frequency carriers: E1-1575.420 MHz, E6-1278.750 MHz, E5a-1176.450 MHz and E5b-1207.140 MHz (see Figure 1). Each of those occupies a different bandwidth: E1-24.552 MHz, E6-40.920 MHz, E5a and E5b-20.460 MHz, respectively. The Galileo frequency bands are allocated for Radio Navigation Satellite Services (RNSS), and in addition to that, the E5a, E5b and E1 bands are also allocated for Aeronautical Radio Navigation Services (ARNS), employed by civil aviation users for dedicated safety-critical applications.

The technical characteristics of specific Galileo service signals are well known, but some parameters, especially for the PRS, for obvious reasons, are classified or even restricted [2]. All the signals are transmitted in the same bands, but with different modulation schemes, so that they can be distinguished. The E1 OS signals are modulated using the Composite Binary Offset Carrier (CBOC), E6 HAS signals (denoted as CS—Commercial Service) use the Binary Phase Shift Keying (BPSK), E5a and E5b signals use the Alternative BOC modulation (AltBOC), and E1 PRS and E6 PRS signals are modulated with the Binary Offset Carrier with cosine phasing (BOCcos) [2]. The technical characteristics of the Galileo Signals in the E1, E6 and E5 bands are presented in Table 1.

## 3. PRS Service Management Authority

A designated Competent PRS Authority (CPA) is required in each EU Member State using PRS or manufacturing PRS technologies. The CPA has an essential role in implementing and managing the PRS across the European Union. In Poland, the CPA has been operating since September 2014 as part of the Ministry of the Interior and Administration, and more specifically, its ICT Department’s Radio Communications and Satellite Systems Unit [3].

The main tasks of CPA Poland include issuing permits for access to the PRS service and for manufacturing of PRS equipment or development of PRS technologies, as well as supervising of PRS users. This organization is also responsible for the supervision of the R&D projects in the PRS area and of the institutions involved in them.

For the management of the PRS signal, there are also two major ground facilities:GSMC (Galileo Security Monitoring Centre) which guarantees that sensitive information related to the use of the PRS service is properly managed and protected, and it also serves as an interface with government entities for requests for cryptographic keys and with the basic components of Galileo for the management of signal messages related to satellites.GCC (Galileo Control Centre), whose main duties include the monitoring and control of the space segment, the processing of data and system assets distributed throughout the world and the support for ground activities prior to the launch of satellites.

The procedure of keying the PRS receivers is presented in a simplified way in Figure 2. The End User submits a PRS Request to the national CPA, which is subsequently transferred to the GSMC and next to the GCC. After validation, the keys are transferred back to the CPA, where the End User terminal is keyed using Black Box Key Management Equipment (BBKME).

It is worth noting that the National Institute of Telecommunications (NIT), with which the authors of the article are affiliated, actively supports CPA Poland in the process of formally launching the Galileo PRS service in Poland. The cooperation includes: (a) joint participation in the international R&D projects, (b) preparation for the national Galileo PRS equipment evaluation measurements and (c) development of the Polish GNSS threats detection system concept (financed by the Chancellery of the Prime Minister and with additional cooperation of the Office of Electronic Communications).

## 4. Coexistence with RNSS Systems and Other Services

All global GNSS systems occupy the same frequency resources, which are allocated to all RNSS services. Over the course of a few recent years, an issue of harmful interference affecting the RNSS receivers and caused by an amateur service (the so-called 23 cm band) has become a source of significant controversy in Europe. Those interference incidents, formally reported by at least two European countries, occurred in the E6 band (1260–1300 MHz), triggering questions on possible actions to protect the RNSS from this undesirable effect.

As specified in the latest, 2020 edition of the ITU-R’s Radio Regulations (RR), the frequency band 1240–1300 MHz is allocated on a primary basis to the radio navigation satellite service RNSS (space–to–Earth and space–to–space). Other services that have co-primary allocations in that band are radiolocation, Earth-exploration (active) and space research (active), but these three are unrelated to the topic of this article. On the other hand, the segments of this band are also allocated—but on a secondary basis—to the amateur service (1240–1300 MHz) and to the amateur satellite service (1260–1270 MHz, via RR footnote 5.282). All frequency allocations mentioned above are worldwide, which means they apply to all three ITU-R Regions [4].

When analyzing the current utilization of the 1240–1300 MHz frequency band by the RNSS, we can conclude it is heavily used by various GNSS systems occupying specific spectrum blocks, including [5]: the European Galileo (1260–1300 MHz), Russian GLONASS (1240–1260 MHz), Chinese Compass (1250–1280 MHz), Japanese QZSS (1260–1300 MHz) and American GPS (some transmissions in the 1215–1240 MHz band may extend above 1240 MHz). In the context of Galileo, it should be noted that the E6 band is essential for the operation of several important services planned in the system, including the HAS and, primarily, the PRS. Consequently, the interference with the RNSS in that band can be a very serious threat for the future development, and operation, of the European GNSS system.

At the same time, the utilization of this band by radio amateurs is also significant [5]. The IARU (International Amateur Radio Union) band plan for Region 1 (Europe, Middle East, Africa and northern Asia) divides the 1240–1300 MHz range into portions to be used by numerous wideband and narrowband amateur applications, including: digital voice, telegraphy, amateur TV (digital or analogue), satellite services and machine-generated modes (MGM). Various applications relevant to that frequency range translate into different types of possible stations, different transmitted power values and/or different sizes/parameters of the utilized antennas. As a whole, it generates a lot of requirements to be fulfilled in order to ensure the smooth coexistence of these two service categories. The number of radio amateurs obviously varies among countries, but it is generally noticeable, e.g., in Poland, there are approximately 13,000 private persons and almost 450 clubs registered with a Category 1 license (maximum allowed output power of 500 W). This is also the factor that needs to be considered when analyzing the compatibility between the amateur services and RNSS.

As mentioned before, the amateur and amateur satellite services have worldwide secondary allocations in the 1240–1300 MHz band. According to the Radio Regulation’s terminology, the secondary status of a service means that the stations of such service [4]:shall not cause harmful interference to stations of primary services to which frequencies are already assigned or to which frequencies may be assigned at a later date;cannot claim protection from harmful interference from stations of a primary service to which frequencies are already assigned or may be assigned at a later date;can claim protection, however, from harmful interference from stations of the same or other secondary service(s) to which frequencies may be assigned at a later date.

Despite these provisions, the interference incidents affecting the RNSS did actually occur. Most notably, there were two cases reported by European countries [5]:Italy: in May/June 2021, several incidents of harmful interference were registered by the Joint Research Center (JRC) of the European Commission in the Varese region. High-end GNSS receivers were affected during the test of a new Galileo High Accuracy Service in the 1260–1300 MHz band. As it turned out, the interference was caused by an FM modulated signal transmitted by an Amateur Radio Repeater. This strong narrow-band emission was received at 1297.3 MHz and characterized by a strong power, more than 40 dB above the noise floor.Germany: an RNSS reference receiver operating in the range 1260–1300 MHz was adversely affected by amateur applications, later identified as amateur TV emissions.

Measurements conducted in other European countries—for example, in Poland—also confirmed that the E6 band is periodically interfered with by amateur applications (see also Section 5.1), so the issue should not be disregarded or treated as incidental.

The reports on these cases were extensively discussed on international forums, including CEPT (European Conference of Postal and Telecommunications Administrations) and ITU-R. As a result of that, the issue of RNSS interference caused by amateur and satellite amateur services in the range 1240–1300 MHz was assigned a dedicated agenda item, 9.1(b), for the upcoming World Radiocommunication Conference WRC-23, which is scheduled to take place in November/December 2023 in the United Arab Emirates. 

At this point, we should note that the World Radiocommunication Conference is an international event held every three to four years, whose job is to review and, if necessary, revise the Radio Regulations, the international treaty governing the use of the radio-frequency spectrum and the geostationary-satellite and non-geostationary-satellite orbits. The very fact the topic of the RNSS interference by the amateur services in the E6 band has been given a separate WRC-23 agenda item shows that this issue is taken very seriously by the international community. The WRC-23 agenda item 9.1(b) will be dedicated to “the review of the amateur service and the amateur-satellite service allocations in the frequency band 1240–1300 MHz to determine if additional measures are required to ensure protection of the radionavigation-satellite (space–to–Earth) service operating in the same band”. The detailed scope of this item covers the following two tasks [6]: (1) to perform a detailed review of the different systems and applications used in the amateur service and amateur-satellite service allocations in the frequency band 1240–1300 MHz and (2) taking into account the results of the above review, to study possible technical and operational measures to ensure the protection of RNSS (space–to–Earth) receivers from the amateur and amateur-satellite services in the frequency band 1240–1300 MHz, without considering the removal of these amateur and amateur-satellite service allocations.

Please note that while the planned actions of the WRC-23 on this matter do include the possible introduction of appropriate measures to protect the RNSS, those measures should (and will) not result in the removal of amateur and amateur-satellite service allocations in the 1240–1300 MHz band. That means that the possible outcome of this WRC-23 agenda item may include specific technical solutions (e.g., revised RNSS protection criteria, new power limits, separation distances for various scenarios, etc.), but other than that, the current allocation scheme for the amateur services in the 1240–1300 MHz range will remain unchanged. 

At the time of this writing (winter 2022), extensive preparatory work for the 9.1(b) agenda item is being carried out worldwide. These activities are conducted by the interested countries (administrations), as well as within international regulatory bodies (e.g., CEPT), and they include technical/legal analysis, simulations, attempts to define possible measures to mitigate the interference issue, etc. The global coordination of this work is handled by the ITU-R’s working parties (WPs), mostly the WP 5A (ITU-R’s Working Party 5A (WP 5A)-Land mobile service, excluding IMT; amateur and amateur-satellite service) with the support of WP 4C (ITU-R’s Working Party 4C–Efficient orbit/spectrum utilization for the mobile-satellite service (MSS) and the radiodetermination-satellite service (RDSS)). Among the expected outputs from those working parties are:The new ITU-R report on the amateur and amateur-satellite services characteristics and usage in the 1240–1300 MHz frequency band,The new ITU-R report containing studies and guidelines regarding the protection of the primary radionavigation–satellite service (space–to–Earth) from the secondary amateur and amateur-satellite service in the frequency band 1240–1300 MHz,Draft of new ITU-R recommendation containing the guidance on technical and operational measures for the use of the frequency band 1240–1300 MHz by the amateur and amateur-satellite service in order to protect the RNSS (space–to–Earth),A consolidated report that will serve as a technical basis for the considerations on the 9.1(b) agenda item during the WRC-23 conference.

Please note that regardless of the progress on the above documents made in the WRC-23 preparatory period, all binding, regulatory decisions will only be made during the conference itself. Therefore, we have to wait till December 2023 to find out what technical measures (if any) will be established with respect to the amateur services in the E6 band in order to efficiently protect the RNSS from harmful interferences.

## 5. Polish Contribution for the PRS

At the end of 2016, the European Union Agency for the Space Programme, EUSPA (at that time, the European GNSS Agency, GSA), announced the Joint Test Activities call, which would result in grants awarded to EU Member States involved in tests and measurement of the Galileo PRS service. Poland took part in two international projects: the PRS Pilot Project for Demonstration (acronym–3PfD) and GNSS Interference Monitoring and Mitigation for End Users–PRS (acronym GIMME PRS). The 3PfD project, conducted in 2018–2022, was coordinated by the Royal Military Academy from Belgium and involved partners from Germany, Finland, Poland and Sweden. The aim of the project was, on the one hand, to help the PRS participants’ CPAs fully comprehend the PRS service and how to interact with the system and, on the other hand, to perform real operational tests of the PRS during dedicated measurement trials and campaigns. One of the most important tasks was to monitor the possible signal interferences across the project partners’ territories. The 3PfD has designated a number of suitable monitoring sites with a high degree of confidence for the purpose of collecting events that would adversely affect the PRS spectrum. Sensors at these sites monitored the GNSS spectrum for an extended period, which made it possible for the consortium to build a waveforms database containing the potential PRS signal interferers. This database and the applied sensors and techniques can now be used by the Member States for better spectral interference management in their respective territories. 

The second project, GIMME, which has been ongoing since 2018, is coordinated by CPA Poland and involves partners from Czech Republic, Finland and Poland. Its main objective is to promote and evaluate the added value of the Galileo PRS in hostile environments where unintentional, as well as intentional, interference with GNSS signals may occur. The conducted field tests will enable validating the PRS in specific use cases and will evaluate the suitability of the PRS to meet the needs of potential users, such as the border control and coast guard, and also will give an opportunity to compare the quality of the PRS signal and its resistance to jamming and spoofing.

The NIT is involved in both of the above-mentioned R&D grants, and also carries out other activities relevant to the Galileo PRS preparatory actions in Poland. Some of them will be presented below.

### 5.1. Galileo PRS Threats Detection

Over the course of the 3PfD project, the Galileo E1 and E6 bands were monitored to detect incidents of intentional and unintentional interference that could potentially degrade the Galileo PRS receiver performance. The data were collected from sensor sites (STRIKE-P) in five participating countries: Belgium, Germany, Finland, Poland and Sweden. All detected events were then classified and assigned to different categories (signal types) with different priority, and subsequently, new records were added to the 3PFD Threat Database. The Polish STRIKE-P station was installed in the headquarters of the future Galileo PRS governmental authorized user in Warsaw (see Figure 3). The monitoring station was composed of two Protector Units for the E1 and E6 bands and a rooftop antenna with a clear sky view. The monitoring station was active for over a year (from 20 August 2020 till 29 September 2021).

The overall statistical analysis of all the events recorded in Poland in the E1 and E6 bands shows that in the E1 band, the interference was observed for less than 2% of the station work time, but in the E6 band, for almost 5% of that time. The average duration of the interference in the E1 band was evenly spread throughout the entire period of the measurement campaign and amounted to a few seconds. Despite the differences in the number of events over time, no correlation was observed with respect to their average duration, and the longest detected interference incidents lasted about 100 s. The situation was different for the E6 band, where the average interference duration value was much greater than in E1. In this case, high mean values (approx. 50 s) were noticed for several months in a row between December 2020 and April 2021 and in July and August 2021. The three longest interferences in the E6 band, lasting 35, 46 and 49 min, occurred on two consecutive days in September 2021. The histogram of the interference duration time in both the E1 and E6 bands in Poland is presented in Figure 4.

It can be easily seen that due to the coexistence of Galileo and amateur services in the E6 band, there is a need to closely monitor this part of the spectrum; some regulatory decisions might also be required, as was mentioned in Section 4.

It is worth noting that during the measurement conducted in the project, cases of the same interference types were detected at the same time in different parts of Europe, which indicates deliberate jamming activity across Europe, probably coming from the space segment of one of the satellite systems. In the face of a growing number of detected incidents and due to the tense international situation, the NIT introduced the concept of the Polish system for GNSS threats detection. The first major component of this system, i.e., the GNSS Galileo Mobile Measurement Platform (Polish acronym: MSP2G) dedicated to monitoring the quality of satellite navigation services and to detecting the threats to GNSS’ integrity and availability, has just been developed in cooperation with the Chancellery of the Prime Minister, which also financed this initiative. Work is currently underway on the development of further components of the national GNSS threats detection system.

### 5.2. GNSS Jamming Tests in a Controlled Laboratory Environment

For the purpose of the jamming test within the 3PfD project, a high-flexibility measurement station has been designed and developed, taking advantage of the NIT’s experience in that field. The tests goal was to evaluate the interference resistance of GNSS receivers in smartphones (as one of the GNSS commercial solutions). To avoid live GNSS signals coming directly to the testing environment and also to prevent jamming signals from leaking outside the laboratory, the isolated measurement location in the basement of the NIT’s office was selected for the purpose of the jamming tests. For the purpose of testing, a high-flexibility measurement station (presented in Figure 5) has been developed, which consists of (the key elements):GNSS Constellation Simulator (Spirent GSS7000),Reference GNSS RTK Receiver (GINTEC M1G2),Vector signal generator (Agilent N5182A),SDR Platform (National Instruments USRP-2945R),Spectrum analyzers (Keysight N9914A, Anritsu MS2721B),Different smartphones from various manufacturers capable of GNSS raw measurements,Computer with the measurement applications.

Live signals from the active GNSS antenna (installed on the building’s rooftop), amplified with a low-noise amplifier (LNA) or artificial signals from a GNSS Constellation Simulator, were combined with the jamming signals and then transmitted by the GNSS passive antenna. In each step, the signals were measured by the spectrum analyzers. A reference position of the rooftop antenna has also been obtained using the GNSS RTK receiver, which facilitated the position determination accuracy tests in the presence of jamming. The campaign was conducted in 2019 and 2020 using different manufacturers’ top-market (at that time) Galileo-compatible smartphones featuring GNSS raw measurements capabilities. In practice, though, due to some compatibility problems with the raw measurements, a dedicated Android Measurement App had to be developed by the NIT on the basis of [7]. This application allowed logging the NMEA data with basic GNSS parameters, satellite status and all the available raw measurements of the GPS and Galileo in the L1/E1 and L5/E5 frequency bands, and to subsequently create RINEX data on their basis. The NIT also developed the PC Measurement Application, which allowed controlling the Android Measurement App on all smartphones via USB connection, to perform simultaneous measurements on all smartphones and to collect the logged data.

During the tests, all the smartphones were switched to Airplane mode in order to avoid the Assisted GNSS mode activation, which could affect the results. After initial tests, some of the smartphones had to be excluded from further stages due to problems with raw data logging. Ultimately, the jamming tests were performed on four types of smartphones: Google Pixel 3 XL, Huawei Mate 20 Pro, Samsung Galaxy S9+ and Sony Xperia XZ3.

For the purpose of the jamming test, we created a large library of interfering signals featuring various interference mechanisms, occupying different bandwidths and exhibiting various time characteristics. After the initial analysis of several variants, it was decided that six general interference signals would be considered in further tests:INT1, Continuous wave (unmodulated carrier)—the single sine wave with constant amplitude and frequency, centered at L1 = 1575.42 MHz;INT2, Three continuous waves—the three sine waves with constant amplitude and frequency, centered at 1574.42 MHz (L1 − 1 MHz), 1575.42 MHz (L1) and 1576.42 MHz (L1 + 1 MHz),INT3, Wideband AWGN signal—the Additive White Gaussian Noise in the 60 MHz bandwidth covering the L1/E1 and G1 GNSS bands,INT4, Narrowband modulated carrier—the QPSK-modulated signal with 4 MHz bandwidth,INT5, Pulse—the rapid increase of the signal amplitude from a baseline value to a higher value, followed by a rapid return to the baseline value, with 1 µs impulse length and different repetition periods in the range of 20 ms to 2 µs,INT6, Sweep (linear chirp)—the constant amplitude signal, in which the frequency increases over 100 µs time, with 5 kHz, 10 kHz or 20 kHz steps in a 4 MHz bandwidth, from 1573.47 (L1 − 2 MHz) to 1577.47 MHz (L1 + 2 MHz).

The first part of the test was performed for the static scenario and the second for the dynamic one. At the beginning of each test, the actual GNSS signal level at the output of the L1/L2 amplifier (in the case of static tests) or at the high-power output of the GNSS Constellation Simulator (in the case of dynamic tests) was measured (with respect to the 4 MHz channel) to obtain the proper C/I (carrier–to–interference ratio) for further jamming tests. The detailed analysis of the collected measurement results was conducted using the RTKPOST application, a part of the RTKLIB library [8]. The receiver position was determined using the Single positioning method in the L1/E1 bands, using only the Galileo or GPS system. On the basis of the static test and for the purpose of the dynamic test, two smartphones with the best and worst resistance to jamming were identified, and three types of the most harmful jamming signals were used. The dynamic scenario was designed in compliance with technical specification 3GPP TS. 25.171 [9], which defines a test case to verify the receiver’s capability to produce GPS measurements or location fixes in a vehicle that is moving, slowing down, accelerating and turning. In both cases (static and dynamic), the detailed statistical analysis was conducted, aimed at the quality of the determined position based on the accuracy and precision measures with respect to the high-accuracy reference RTK position of the rooftop antenna in the case of static tests and the reference route for dynamic tests. 

For further comparative analysis, the R95 measure (radius of the circle with the center in the actual position, which covers 95% of the measurements) was chosen as one of the most frequently used [10,11]. The Dilution of Precision (DOP) parameters were also observed. On the basis of the measurement results, the so-called C/I cut-off level and C/I degradation level were determined for most types of interference:The C/I cut-off level was defined as the highest C/I ratio, for which the transition from the “3D fix” state to the “no-fix” state was observed.The C/I degradation level was the highest C/I ratio, for which the accuracy (related to the reference RTK position in the case of static tests and the reference route for dynamic tests) starts to exceed the value of 15 m.

Table 2 and Table 3 present the interferences’ impact levels for the smartphones tested in the Galileo and GPS scenarios for the static and dynamic tests, respectively.

INT1 (Continuous Wave), the simplest interfering signal, had a minor impact on most of the tested smartphones, especially for the GPS scenario. As the interference level was increased, the corresponding degradation of the decoded signal quality in the receiver (denoted as the SNR level) could be observed. The degradation rate was more rapid for the GPS than for Galileo. Further, the DOPs values were affected, which could be seen as gradual or abrupt deterioration or significant fluctuations. The most significant impact was noticed for the Samsung smartphone (for both Galileo and GPS), in which case the interferer caused almost complete degradation of the received signal from the beginning of the jamming signal occurrence. The Google and Sony smartphones were only affected when working with the GPS. In the presence of the INT1, no smartphones, except for Samsung, lost their ability to determine their position using Galileo signals. In fact, the Huawei was so resilient that even at high levels of INT1, it was still able to determine its position.

INT2 (Three Continuous Waves), also one of the simplest interfering signals, had a moderate impact on all of the tested smartphones in both the static and dynamic tests. In the static tests, the degradation of the decoded signal quality in the receiver (denoted as the SNR level) could be observed as the interference level was increased. The degradation rate was equal for GPS and Galileo. Further, the DOPs values were adversely affected. The most significant impact was noticed for the Sony smartphone in the Galileo mode, where the DOPs degradation was observed from the beginning of the jamming signal occurrence and where the inability to determine the position occurred the earliest. All the smartphones, except for Huawei, eventually lost their positioning ability while facing the INT2 interferer. Huawei was partially resistant to this jamming signal and, for Galileo, it was not affected at all. In the GPS scenario, though, at high levels of interference, its positioning accuracy decreased rapidly. In the case of the dynamic tests, the most significant impact was noticed for the Sony in the Galileo mode, where in all the scenarios, the smartphone lost its ability to determine its position. In the GPS mode, it was capable of positioning, but only at the lowest tested interference level. In the case of Huawei and Galileo, the accuracy of the determined position was slightly decreased at high INT2 levels, and in the GPS mode, the position accuracy rapidly decreased after exceeding the CIR level of −55 dB, which was also observed in the static tests.

INT3 (Wideband AWGN noise L1/E1 60 MHz), one of the most popular jamming signals, had a significant impact on all the tested smartphones in both the static and dynamic tests. In the case of the static test, the influence of the increasing interference level added to the GNSS signals was observed as the degradation of the quality of the signal decoded in the receiver for each satellite (denoted as the SNR level). The degradation rate was equal for GPS and Galileo. Further, the DOPs values were disturbed and deteriorated. The only exception was the Huawei in the Galileo scenario, which was almost unaffected. The most harmful impact was noticed for the Sony smartphone in both the Galileo and GPS cases, where INT3 caused the DOPs’ degradation from the very beginning of the jamming signal occurrence. All the smartphones, except for Huawei, lost their positioning ability. Huawei was partially resilient to that jamming signal, and for the Galileo mode, it was almost unaffected; but in the GPS mode, at high levels of the INT3 signal, its ability to determine its position was lost. In the case of the dynamic tests, the most harmful impact was noticed for the Sony in both the Galileo and GPS scenarios, where the positioning accuracy rapidly decreased after exceeding the CIR level of −30 dB; for higher CIR levels, the smartphone gradually lost its positioning ability altogether. Huawei in the Galileo mode remained almost unaffected in the static tests, but during the dynamic tests, its positioning accuracy decreased as the interference increased, and eventually it lost its positioning ability in both the Galileo and GPS scenarios. The most significant impact of this interference can be observed for the Sony smartphone in both the Galileo and GPS modes.

INT4 (Narrowband modulated carrier 4 MHz), one of the sophisticated jamming signals, had a significant impact on all of the tested smartphones. As the interference level was increased, the corresponding degradation of the decoded signal quality in the receiver (denoted as the SNR level) could be observed. The degradation rate was equal for GPS and Galileo. Further, the DOPs values were adversely affected. The most harmful impact was noticed for the Sony smartphone in both the Galileo and GPS scenarios, where the DOPs degradation was visible from the beginning of the jamming signal occurrence. On the other hand, for Samsung in the GPS mode, the increase of the DOPs values was much slower and smoother than in other smartphones, and after reaching the C/I cut-off value, they just dropped down to zero, denoting no fix. All the smartphones eventually lost their positioning capabilities, but the Huawei (in the Galileo mode) recovered this ability later, despite even higher interference levels.

INT5 (Pulse), one of the sophisticated jamming signals, had a minor impact on the Huawei and Samsung smartphones, a moderate impact on the Google smartphone and a significant impact on the Sony smartphone. In some scenarios, the presence of the interference caused the degradation of the decoded signal quality in the receiver for each satellite (denoted as the SNR level), which was particularly noticeable for the 2 µs pulse repetition period; for the Sony, the “no fix” state was even observed. The pulse jamming signal also caused the disturbances and increased the DOPs values. The most harmful impact was noticed for the Sony in both the Galileo and GPS scenarios, causing DOPs degradation for pulse repetition periods lower than 2 ms. In the presence of INT5, Sony lost positioning capability on several occasions, but in the Galileo mode, it happened more frequently than for the GPS mode. On the other hand, the Huawei and Samsung were mostly unaffected by INT5 in the Galileo mode, and for the GPS, their accuracy degradation was only observed for a few pulse signal repetition period cases. 

INT6 (Sweep), also one of the complex jamming signals, had a severe impact on all the tested smartphones in both the static and dynamic tests. In the case of the static tests, the presence of the interference caused a significant degradation of the quality of the signal decoded in the receiver for each satellite (denoted as the SNR level). On some occasions, after the application of the INT6 interferer, the useful signals were completely lost, and the DOPs values dropped down to zero, denoting no fix. The most harmful impact was noticed for all the smartphones in the GPS mode, where the positioning capability was lost for most swept signal configurations. In the Galileo mode, Google and Sony lost their positioning capability in most of the INT6 jamming cases. On the other hand, Samsung and Huawei in the Galileo mode were still able to determine their position, despite the presence of interference, and the accuracy degradation was noticeable only for Samsung. In the case of the dynamic test, the results were different than in the static one. Huawei almost completely lost its positioning ability in both the Galileo and GPS scenarios, and for Sony, the observed degradation of the positioning accuracy was rather moderate.

Summing up, the simplest possible interfering signals, INT1 (Continuous Wave) and INT2 (Three Continuous Waves), had a minor to moderate impact on all of the tested smartphones. More sophisticated jamming signals, INT3 (Wideband AWGN noise L1/E1 60 MHz) and INT4 (Narrowband modulated carrier 4 MHz), had a significant impact on all of the tested smartphones. The complex jamming signals, INT5 (Pulse), had a wide range of impact levels on various smartphones, from minor to significant, and INT6 (Sweep) had a severe impact on all tested smartphones. 

In most cases, the Sony smartphone was the most vulnerable to the jamming signals, and in many cases, the DOPs degradation was visible from the beginning of the jamming signal occurrence. Moreover, in most cases, the Sony had the highest C/I cut-off and degradation levels. On the other hand, the best resistance to the jamming signals during static tests was featured by the Huawei smartphone, especially for the Galileo case. It can be concluded that the Broadcom BCM47755 chip that the Huawei Mate 20 Pro was equipped with has, as stated by its manufacturer, a high immunity to interference and jamming [12]. On the other hand, in the case of the dynamic tests, and especially in the presence of one of the most sophisticated jamming signals, the sweep, the resistance of the Huawei smartphone was worse than the most vulnerable smartphone in the static tests, i.e., Sony.

It is worth noting that position determination algorithms in smartphones normally use all the available GNSS signals, and the various sensors that the smartphones might be equipped with (e.g., the accelerometer, gyroscope and magnetometer) may also support the inertial navigation system (INS). The performed jamming tests, conducted in a static scenario in the laboratory environment, demonstrate that in many cases, the GNSS smartphones receivers were still able to determine the position, as indicated by the saved NMEA logs. Only in the case of INT3, i.e., the wideband AWGN noise covering all 60 MHz of the L1/E1 frequency band, which affects all GNSS systems, did the smartphone receivers lose (in most cases) their ability to determine their position.

## 6. Conclusions for Future PRS Tests

In recent years, threats to the integrity and availability of GNSS systems have increased. The Galileo Public Regulated Service (PRS), dedicated for authorized European governmental users, is one of the most promising solutions, being more resilient and robust than the initial GNSS open services. The Full Operational Capability phase of the PRS service is planned for 2023 [13], but the monitoring stations across Europe, including in Poland, are already reporting more and more cases of unintentional or deliberate activities in the GNSS frequency bands. Moreover, the coexistence of the Radio Navigation Satellite Services (RNSS) with the amateur service within the E6 band has become a source of significant controversy in Europe. Until the upcoming World Radiocommunication Conference, WRC-23, neither technical measures nor regulatory decisions will be possible or made. Due to the importance of the increasing security risk for GNSS, it is essential to act now.

The experience gained by the National Institute of Telecommunication (NIT) during the jamming tests discussed in the article (selected jamming signals library, design and development of the high-flexibility measurement station and development of measurement methodology) can be used for the future tests aimed at the Galileo PRS service’s resistance to intentional interference. The proposed measurement station can be easily adjusted for the purpose of the future measurements, both in the controlled laboratory environment and in outdoor dedicated test sites. It also can be adopted for mobile applications. The jamming signal library and the static and dynamic scenarios can be easily extended to create various sophisticated tests.

The results presented in the article showed that the resistance to various jamming signals depends on the manufacturer of the GNSS receiver (both the GNSS chipset and the smartphone). It can, therefore, be expected that a similar effect might be observed with different manufactures of the PRS receivers. The pre-operational PRS receivers are now available, and in the near future, it can be expected that some EU Member States will develop their own PRS receivers to support national authorities and organizations responsible for security tasks and authorized and governmental users. The experience gained by the NIT is also used to support CPA Poland in the process of formally launching the Galileo PRS service in Poland, and it constitutes the basis for the preparation for the national Galileo PRS equipment evaluation measurements laboratory for CPA Poland purposes.

The future measurement can also evaluate the added value of Galileo PRS over other GNSS services and give an opportunity to compare its quality and resistance to jamming and spoofing with other services. The PRS field tests will also enable PRS validation in specific use cases, as well as assess the suitability of the PRS to meet the needs of potential users in hostile environments, in which unintentional, as well as intentional, GNSS interference may occur. 

Nowadays, the risk of deliberate interference with GNSS signals is substantial and growing, and may also be used to support military operations [14,15,16]. Such actions have become an inseparable element of modern warfare and are used to disturb the navigation systems of surveillance drones and targeting missiles, as well as to destabilize (cellular) radio communication networks that use GNSS signals, e.g., for synchronization purposes. As GNSS signal jamming has now become a part of military offensives, the role of the Galileo PRS service, resistant to typical spoofing and jamming attacks, is of great importance for the safety and security of the European Union. It can, therefore, be expected that the risk of disrupting the Galileo PRS will be even greater than for the initial GNSS Open Services. The methods of GNSS OS signal jamming and spoofing are widely known and available nowadays [17], but due to the resilience and robustness of the Galileo PRS service, attacking its signals will definitely be much more difficult. 

The increasing risk of intentional interference events that could affect GNSS services and the receivers’ performance is driving the necessity to develop national systems for GNSS threats detection. In the case of the Polish system, the low-level system monitor will be based on the time and spectral analysis of the frequency bands dedicated to GNSS systems and detection of unexpected jamming signals. On the other hand, the high-level system monitor will be based on monitoring the quality, integrity and availability of specific GNSS services in order to identify the presence of spoofing signals. The first components of such a system (equipped with sophisticated laboratory measurement equipment)—the GNSS Galileo Mobile Measurement System (Polish acronym: MSP2G)—has just been developed by the NIT in cooperation with the Chancellery of the Prime Minister, which financed the project. On the basis of the experience gained during the MSP2G functional tests, further components (probes) of the national GNSS threats detection system will be developed, utilizing software-defined radio platforms dedicated to various specific applications.

## Figures and Tables

**Figure 1 sensors-23-01770-f001:**
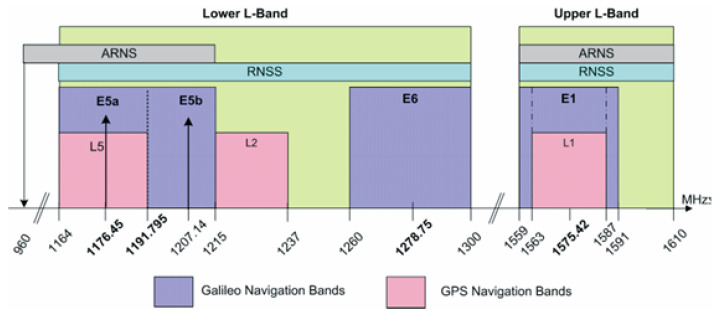
Galileo Frequency Plan [2].

**Figure 2 sensors-23-01770-f002:**
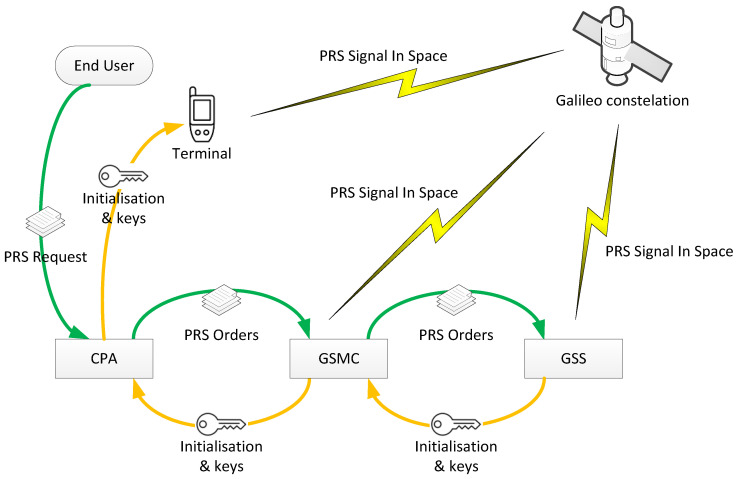
PRS keying procedure.

**Figure 3 sensors-23-01770-f003:**
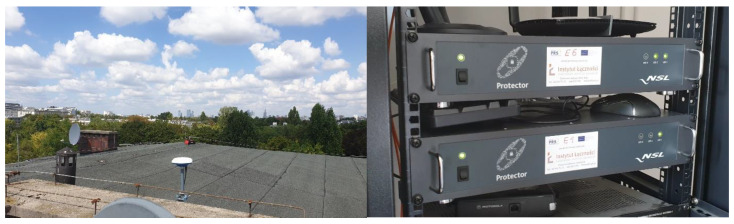
The Polish STRIKE-P monitoring station.

**Figure 4 sensors-23-01770-f004:**
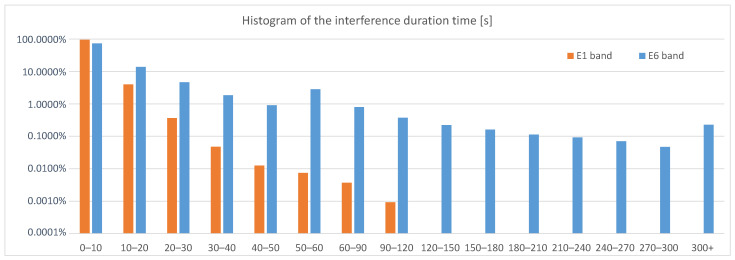
Histogram of the detected interference duration time in the E1 and E6 bands in Poland.

**Figure 5 sensors-23-01770-f005:**
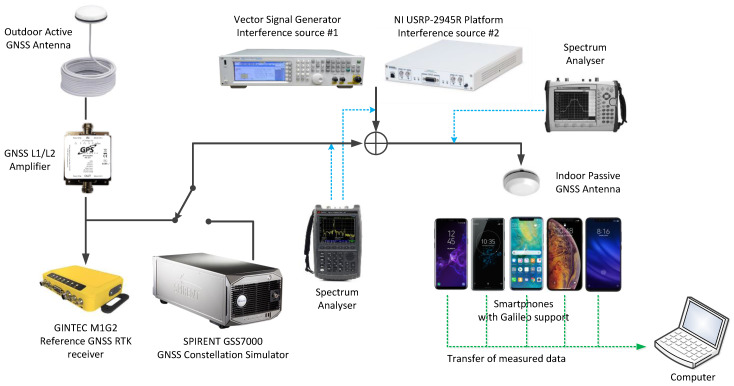
Jamming test measurement station.

**Table 1 sensors-23-01770-t001:** The Galileo signal technical characteristics in different bands.

Frequency Band	E1		E6		E5a & E5b
Service Name	OS data/pilot	PRS	CS data/pilot	PRS	E5a data/pilot/ E5b data/pilot
Center frequency	1575.42 MHz		1278.75 MHz		1191.795 MHz
Access technique	CDMA		CDMA		CDMA
Spreading modulation	CBOC(6, 1, 1/11)	BOC_cos_(15, 2.5)	BPSK(5)	BOC_cos_(10, 5)	AltBOC(15, 10)
Subcarrier frequency	1.023 MHz and 6.138 MHz	13.345 MHz	-	10.23 MHz	15.345 MHz
Code-frequency	1.023 MHz	2.5575 MHz	5.115 MHz		10.23 MHz

**Table 2 sensors-23-01770-t002:** Impact level of various interferences for static jamming tests.

		Google		Huawei		Samsung		Sony	
		GAL	GPS	GAL	GPS	GAL	GPS	GAL	GPS
INT1-Continuous Wave									
Impact Level		Low	High	Low	Low	Medium	High	Low	Medium
C/I Cut-off Level [dB]		N/D	−34	N/D	N/D	−29	−36	N/D	−36
C/I Degradation Level [dB]		−45	−28	N/D	N/D	−23	−20	N/D	−30
	INT2–Three Continuous Waves								
	Impact Level	High	High	Low	High	Medium	High	High	Medium
	C/I Cut-off Level [dB]	−37	−38	N/D	N/D	−36	−36	−3	−40
	C/I Degradation Level [dB]	−12	−32	N/D	−46	−17	−19	4	−33
INT3-Wideband AWGN Noise L1/E1 60 MHz									
Impact Level		High	High	Low	High	Medium	High	High	High
C/I Cut-off Level [dB]		−35	−37	N/D	−39	−32	−37	−31	−36
C/I Degradation Level [dB]		−22	3	−7	−22	−20	−24	2	3
INT4-Narrowband Modulated Carrier 4 MHz									
Impact Level		High	High	Medium	High	Medium	Medium	High	High
C/I Cut-off Level [dB]		−28	−29	−28	−37	−35	−35	−28	−30
C/I Degradation Level [dB]		8	−9	−13	−32	−20	−23	−14	0
INT5–Pulse									
PULSE 1 µs/20 ms		Low	Low	Low	Low	Low	Low	Medium	Medium
PULSE 1 µs/2 ms		Low	Low	Low	High	Low	Low	High	Low
PULSE 1 µs/200 µs		Low	High	Low	Low	Low	Low	High	High
PULSE 1 µs/20 µs		Low	High	Low	Low	Low	Medium	Medium	High
PULSE 1 µs/2 µs		High	High	Low	Low	Low	Medium	High	High
INT6-Sweep									
Sweep 5 kHz/100 µs		High	High	Low	High	Medium	High	High	High
Sweep 10 kHz/100 µs		High	High	Low	High	Medium	High	High	High
Sweep 20 kHz/100 µs		High	High	Low	Low	Medium	High	High	High

N/D—specific level not determined during measurements.

**Table 3 sensors-23-01770-t003:** Impact level of various interferences for dynamic jamming tests.

	Google		Huawei		Samsung		Sony	
	GAL	GPS	GAL	GPS	GAL	GPS	GAL	GPS
INT2–Three Continuous Waves								
Impact Level	N/A	N/A	Low	Medium	N/A	N/A	High	High
C/I Cut-off Level [dB]	N/A	N/A	N/D	N/D	N/A	N/A	−40	−45
C/I Degradation Level [dB]	N/A	N/A	−45	−55	N/A	N/A	N/D	N/D
INT3-Wideband AWGN Noise L1/E1 60 MHz								
Impact Level	N/A	N/A	Medium	High	N/A	N/A	High	High
C/I Cut-off Level [dB]	N/A	N/A	−38	−38	N/A	N/A	−36	−36
C/I Degradation level [dB]	N/A	N/A	−30	−35	N/A	N/A	−35	−30
INT6-Sweep								
Sweep 5 kHz/100 µs	N/A	N/A	High	High	N/A	N/A	Medium	Medium
Sweep 10 kHz/100 µs	N/A	N/A	High	High	N/A	N/A	Medium	Low
Sweep 20 kHz/100 µs	N/A	N/A	High	High	N/A	N/A	Medium	Low

N/D—specific level not determined during measurements. N/A—specific scenario not tested.

## Data Availability

The measurement data presented in the manuscript are not publicly available.

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
