# Peer review of "Preparations for Galileo PRS in Poland"

_sensors, 2023, doi:10.3390/s23041770_

Round 1

Reviewer 1 Report

This manuscript gives a very detailed statement of the services of Galileo, especially for the PRS. They also mentioned the contributions of PRS Poland in developing measuring stations and spoofing/jamming tests. The literature review of this manuscript is complete while the results are comparable weak. As this study is focused on the security aspects of PRS service, I think the authors should add some experiments about the navigation or anti-spoofing/jamming performance of PRS.

Author Response

Response to Reviewer 1 Comments

Point 1: This manuscript gives a very detailed statement of the services of Galileo, especially for the PRS. They also mentioned the contributions of PRS Poland in developing measuring stations and spoofing/jamming tests. The literature review of this manuscript is complete while the results are comparable weak. As this study is focused on the security aspects of PRS service, I think the authors should add some experiments about the navigation or anti-spoofing/jamming performance of PRS.

Response 1:

First of all, the authors would like to thank the Reviewer for the comments.

The PRS experiments mentioned by the Reviewer have not been carried out yet, but the experience gained by the authors during the GNSS Open Services jamming tests, will definitely be used for the Galileo PRS tests scheduled for 2023. Currently the integration of the PRS receiver with the measurement station is in progress and the measurement methodology is also being adjusted. At this moment, the authors are unable to determine whether it will be possible to publish the PRS measurement results after their completion - it will need to be additionally discussed with the CPA Poland and EUSPA. Due to the European security regulations, some of the results of these Galileo PRS jamming/spoofing tests will probably be classified with the EU restricted or higher clause.

Reviewer 2 Report

This article briefly introduced the relevant situation of Galileo PRS in the EU, especially the two projects in which Poland participated. Therefore, the definition of this paper is neither a general review article nor a research paper, which is like a summary of an engineering project. Therefore, unclear positioning is the biggest problem of this paper. If it is defined as an overview article, the history, current situation, problems and future development of PRS need to be analyzed in detail.

Author Response

Response to Reviewer 2 Comments

Point 1: This article briefly introduced the relevant situation of Galileo PRS in the EU, especially the two projects in which Poland participated. Therefore, the definition of this paper is neither a general review article nor a research paper, which is like a summary of an engineering project. Therefore, unclear positioning is the biggest problem of this paper. If it is defined as an overview article, the history, current situation, problems and future development of PRS need to be analyzed in detail.

Response 1:

First of all, the authors would like to thank the Reviewer for the comments.

With respect to the Reviewer’s comments, the authors would like to present the following point of view:

The article is dedicated to the issue of the GNSS systems’ resistance to the integrity and availability threats. This is definitely a topic of growing importance and we believe it should be of interest for the readers of the Sensors Journal.

In the authors opinion, the selected paper type – “article”, is appropriate, given the contents of the manuscript. The authors present the Polish contribution to the Galileo PRS preparatory actions and in their opinion, the paper meets the attributes of overview article: it discusses the Galileo PRS service signal plan, the PRS service management matters in EU Member States and the important problems of the GNSS E6 band collocation with the amateur services, which will be considered during the WRC-23 conference under a separate agenda item 9.1(b). The paper also discusses the increasing GNSS security risk due to unintentional and deliberate interference (attacks), which has gotten significantly worse since the outbreak of the war in Ukraine. The authors present a significant set of statistical data obtained in Poland and their experience gained during the GNSS Open Services jamming tests (within one of R&D projects), which can and will be used for the future Galileo PRS tests which are scheduled for 2023.  Currently the integration of the PRS receiver with the measurement station is in progress and the measurement methodology is also being adjusted. At this moment, the authors are unable to determine whether it will be possible to publish the PRS measurement results after their completion - it will need to be additionally discussed with the CPA Poland and EUSPA. Due to the European security regulations, some of the results of these Galileo PRS jamming/spoofing tests will probably be classified with the EU restricted or higher clause.

All the elements mentioned above account for a “wide picture” of the discussed topic (the background, formal issues regarding the PRS, regulatory issues to be discussed at WRC-23, results of the measurements and their analysis, etc.), and consequently, we believe the classification of the manuscript as an article was justified.

Reviewer 3 Report

The paper is interesting and relevant for sure. I have only two minor questions/suggestions for improvement:

Section 5.1. Statistics on the interference time is provided, but was this interference meaningful or not?

Line 368 and further: How these exact interference signals were chosen? Are they typical for radio amateur or deliberate enemy interference? Does typical anti-drone equipment, which is extensively used in Ukraine right now, utilize any of these signals (not sure this information is available though)?

Author Response

Response to Reviewer 3 Comments

Point 1: Section 5.1. Statistics on the interference time is provided, but was this interference meaningful or not?

Response 1:

First of all, the authors would like to thank the Reviewer for the comments.

The authors’ intention with respect to the statistics presented in the Section 5.1 was to show the scale of the problem, therefore the histogram depicted in fig. 4 presents all detected interference regardless of their priority.

Point 2: Line 368 and further: How these exact interference signals were chosen? Are they typical for radio amateur or deliberate enemy interference? Does typical anti-drone equipment, which is extensively used in Ukraine right now, utilize any of these signals (not sure this information is available though)?

Response 2:

The interfering signals presented in the article have been chosen on the basis of the  authors’ knowledge and experience gained during different R&D projects. These interfering signals represent: the basic signals (INT1, INT3 – one or three unmodulated carriers), noise in the channel (INT3 – AWGN), typical data transmission signals (INT4 - QPSK modulated signal), and some more complex signals (INT5 - Pulse and INT6 – Sweep) which might be generated using simple jammers or radio amateur devices, but they also might represent part of some coded transmission.

The authors do not know exactly which signals are used for jamming and spoofing GPS signals in Ukraine by the Russian forces – for obvious reasons such information is not publicly available.

Round 2

Reviewer 2 Report

Accept after minor revision。

"GALILEO" in the title should be "Galileo".

Author Response

Response to Reviewer 2 Comments

Point 1: "GALILEO" in the title should be "Galileo".

Response 1:

The authors once again would like to thank the Reviewer for the comments.

The proposed change of the manuscript title has been made.
